# TRPC6 Deficiency Attenuates Mitochondrial and Cardiac Dysfunction in Heart Failure with Preserved Ejection Fraction Induced by High-Fat Diet Plus L-NAME

**DOI:** 10.3390/ijms26199383

**Published:** 2025-09-25

**Authors:** Xuan Li, Yiling Fu, Xuemei Dai, Jussara M. do Carmo, Alexandre A. da Silva, Alan J. Mouton, Ana C. M. Omoto, Robert W. Spitz, Lucas Wang, John E. Hall, Zhen Wang

**Affiliations:** Department of Physiology and Biophysics, Mississippi Center for Obesity Research, Cardiorenal and Metabolic Diseases Research Center, University of Mississippi Medical Center, Jackson, MS 39216, USA; xli3@umc.edu (X.L.); yfu@umc.edu (Y.F.); xdai2@umc.edu (X.D.); jdocarmo@umc.edu (J.M.d.C.); asilva@umc.edu (A.A.d.S.); amouton@umc.edu (A.J.M.); aomoto@umc.edu (A.C.M.O.); rspitz@umc.edu (R.W.S.); lucaswang421@gmail.com (L.W.); jehall@umc.edu (J.E.H.)

**Keywords:** diastolic dysfunction, obesity, hypertension, exercise capacity, dobutamine test

## Abstract

Transient receptor potential canonical channel type 6 (TRPC6), a non-selective cation channel that mediates Ca^2+^ influx, is expressed in the heart and implicated in pathological cardiac hypertrophy. However, the role of TRPC6 in regulating cardiac mitochondrial metabolism and contributing to development of HFpEF remains unclear. We examined whether TRPC6 deficiency prevents mitochondrial dysfunction and offers cardiac protection in a mouse model of HFpEF induced by high-fat diet (HFD) for 12 weeks combined with L-NAME administration during the final 8 weeks in TRPC6 knockout (KO) and wild-type (WT) control mice. Cardiac systolic and diastolic functions were assessed at baseline, 4 and 8 weeks after HFD+L-NAME. Dobutamine-induced stress test and treadmill exercise test were performed at the end of the protocol to evaluate cardiac reserve capacity and exercise tolerance. Mitochondrial oxygen consumption rate (OCR) and mitochondrial-derived reactive oxygen species (ROS) generation were examined in isolated cardiac fibers. WT mice subjected to HFD+L-NAME developed cardiac hypertrophy, diastolic dysfunction, and exercise intolerance, whereas TRPC6 KO mice, under the same conditions, maintained preserved diastolic function, exercise tolerance, and cardiac reserve. We also observed increased TRPC6 in mitochondria, as well as caspase-9 activation and impaired mitochondrial respiration in WT mice. In contrast, TRPC6 KO mice exhibited preserved mitochondrial OCR and attenuated mitochondrial ROS generation. In summary, TRPC6 deficiency prevents the development of HFpEF by mitigating diastolic dysfunction, preserving cardiac reserve capacity, and attenuating mitochondrial dysfunction.

## 1. Introduction

Heart failure with preserved ejection fraction (HFpEF) represents a significant unmet need in cardiovascular medicine, accounting for >500,000 annual hospitalizations in the US and a 5-year mortality of ~50% [1,2,3,4,5]. The prevalence of HFpEF is increasing and becoming the dominant form of heart failure (HF), primarily due to rising rates of obesity and hypertension (HTN) [1,6,7]. HFpEF is characterized by diastolic dysfunction due to impaired ventricular relaxation and increased left ventricular filling pressures. Medications that improve survival and clinical outcomes in HF with reduced ejection fraction (HFrEF), such as ACE inhibitors (ACEI) and β-blockers, generally offer minimal benefit to patients with HFpEF [8,9,10]. Therefore, further understanding the pathogenesis and molecular mechanisms of HFpEF is crucial for identifying novel therapies to protect the heart, particularly for patients with metabolic syndrome and HTN during the early phase of HFpEF.

Metabolic abnormalities and mitochondrial dysfunction are critical contributors to the pathogenesis of HFpEF, impairing myocardial energetics and exacerbating systemic and cardiac metabolic stress [11,12,13]. Studies from our group and others have demonstrated that excess activation of transient receptor potential canonical channel type 6 (TRPC6), a non-voltage-gated cation channel, contributes to mitochondrial dysfunction in the kidney in response to metabolic stress [14,15]. When we examined the potential role of TRPC6 in mitochondrial dysfunction during HFpEF, our initial studies discovered that TRPC6 was expressed in purified mitochondria isolated from the heart (Appendix A). Given that TRPC6 functions as a channel mediating Ca^2+^ influx into the cytoplasm, we further investigated whether mitochondrial expression of TRPC6 may increase with development of HFpEF and play a role in causing cardiac mitochondrial dysfunction. Previous studies suggest that TRPC6 overactivation in the heart may contribute to pro-fibrotic signaling in animal models of pressure overload-induced HF, primarily through the calcineurin–NFAT pathway, which upregulates the expression of fibrosis-related genes [16,17,18,19]. However, to our knowledge, the role of TRPC6 in contributing to mitochondrial and cardiac dysfunction in HFpEF has not been previously investigated.

In this study, we employed a mouse model of HFpEF that combines obesity and HTN by feeding animals a high-fat diet (HFD) and inducing endothelial dysfunction and HTN by inhibiting nitric oxide production using L-NAME (N-nitroarginine methyl ester) [12,20,21]. This model replicates several key characteristics of HFpEF in humans, including endothelial dysfunction, high blood pressure (BP), and early diastolic dysfunction. Using a TRPC6 knockout (KO) model allowed us to comprehensively investigate in vivo physiological changes, in vitro molecular analyses, and genetic targeting of TRPC6 as a potential contributor to the development of HFpEF. Our results suggest that TRPC6 deficiency attenuates mitochondrial dysfunction and prevents the development of HFpEF induced by obesity, endothelial dysfunction, and HTN. Thus, TRPC6 may represent a potential therapeutic target for improving cardiac function in patients with HFpEF.

## 2. Results

### 2.1. Metabolic Profiles and Mean Arterial Pressure Before and After Administration of HFD+L-NAME in WT and TRPC6 KO Mice

In RD-fed WT and TRPC6 KO mice, blood glucose and body weight did not change significantly from the baseline. In contrast, both parameters increased significantly in TRPC6 KO and WT mice after 12 weeks of HFD+L-NAME treatment compared to baseline (Table 1). At baseline, mean arterial pressure (MAP) and HR were comparable between WT and TRPC6 KO mice. However, MAP increased significantly by 17 mmHg in both groups after HFD+L-NAME treatment (Table 1). These findings indicate that our HFpEF mouse model exhibited moderate obesity (20–30% increase in body weight) and mild HTN (~120 mmHg), which are recognized as major risk factors for the development of HFpEF.

### 2.2. Effects of HFD+L-NAME on Cardiac Systolic Function in WT and TRPC6 KO Mice

Overall, there were no significant differences in cardiac systolic function as assessed by stroke volume (SV), cardiac output (CO), ejection fraction (EF), and fractional shortening (FS) at baseline, 26 and 30 weeks of age in WT and TRPC6 KO mice that were fed RD, suggesting TRPC6 deficiency did not affect normal cardiac function (Figure 1A–D). When mice received combined HFD+L-NAME treatment, their SV, CO, EF, and FS remained unchanged in WT and TRPC6 KO mice compared to baseline (Figure 1A–D). Representative echocardiographic images for each group are shown in Appendix A.

### 2.3. Effects of HFD+L-NAME on Cardiac Diastolic Function in WT and TRPC6 KO Mice

We assessed cardiac diastolic function in all groups, including isovolumic relaxation time (IVRT), myocardial performance index (MPI), and E/e’. In WT mice, HFD+L-NAME treatment significantly increased IVRT compared to baseline; however, this effect was not observed in the TRPC6 KO mice (Figure 1E). Additionally, WT mice on HFD+L-NAME showed a gradual increase in MPI over time, whereas no significant changes were detected in the TRPC6 KO HFD+L-NAME group (Figure 1F). Although there was a trend toward increased E/e’ in WT mice on HFD+L-NAME, the changes did not reach statistical significance (Figure 1G). Overall, diastolic function was similar in WT and TRPC6 KO mice fed an RD and remained normal throughout the duration of the protocol. However, mild diastolic dysfunction developed in WT mice after 8 weeks of HFD+L-NAME treatment, whereas TRPC6 KO mice appeared to be protected.

### 2.4. HFD+L-NAME Induced Left Ventricular Hypertrophy in WT and TRPC6 Mice, but Did Not Cause Significant Fibrosis

The thickness of left ventricular anterior (LVAW; s and LVAW; d) and posterior wall (LVPW; s and LVPW; d) at systole and diastole are commonly used to indicate the extent of left ventricular hypertrophy. WT and TRPC6 KO mice fed with RD showed no significant differences in the thickness of left ventricular walls at systole and diastole between different time points. For WT and TRPC6 KO mice treated with HFD+L-NAME, we did not observe significant changes in the thickness of anterior and posterior walls during systole at each time point (Figure 2A,C). However, for LVAW during diastole, we found a significant increase in WT and TRPC6 KO mice after 8 weeks of HFD+L-NAME (Figure 2B) compared to their baseline. For LVPW during diastole, a substantial increase from baseline can be observed in WT after 8 weeks of HFD+L-NAME but not in TRPC6 mice (Figure 2D). Heart weight was also measured at the end of the protocol, and results showed a significant increase in WT and TRPC6 KO mice treated with HFD+L-NAME compared to those fed an RD (Table 1).

Using Masson’s trichrome staining, we also evaluated cardiac fibrosis in WT and TRPC6 KO mice on RD or HFD+L-NAME (Figure 2E–H). There was a trend for an increase in collagen staining in WT mice receiving HFD+L-NAME compared to WT RD control; however, this change was not statistically significant (Figure 2I). Similarly, no significant differences were observed in TRPC6 KO mice treated with RD or HFD+L-NAME (Figure 2I). These results indicate that while HFD+L-NAME induced left ventricular hypertrophy in WT and TRPC6 KO mice, ventricular fibrotic remodeling appears much less pronounced.

### 2.5. Dobutamine Stress Test Suggested Reduced Cardiac Reserve Capacity in WT but Not TRPC6 KO Mice After HFD+L-NAME

To further investigate the impact of HFD+L-NAME on cardiac function, we conducted a dobutamine stress test and used echocardiography to assess cardiac reserve capacity in WT and TRPC6 KO mice. In RD-fed WT and TRPC6 KO mice, dobutamine injection caused an increase in HR and EF beginning within 1 min post-injection, reaching their peak at 5 min, with no further changes at 10 min (Figure 3A,B). Dobutamine did not significantly alter SV and CO in either group compared to their baseline (Figure 3C,D). In WT and TRPC6 KO mice, left ventricular end systolic volume (LVESV) (Figure 3E) was significantly reduced by more than 80%, and left ventricular end diastolic volume (LVEDV) (Figure 3F) was slightly but significantly reduced at 10 min post-dobutamine injection.

In WT and TRPC6 KO mice treated with HFD+L-NAME, HR, EF, and SV, changes were similar to those fed RD (Figure 3G–I). CO in WT HFD+L-NAME mice had no significant changes following dobutamine injection. However, CO in TRPC6 KO HFD+L-NAME mice was significantly higher at 5 and 10 min post-injection compared to their baseline (Figure 3J). The LVESV and LVEDV measurements showed that end-systolic volume significantly decreased after dobutamine injection in WT and TRPC6 KO mice, regardless of RD or HFD+L-NAME treatment (Figure 3K). However, end-diastolic volume was significantly reduced in WT mice but unchanged in TRPC6 KO mice (Figure 3L), suggesting impaired diastolic relaxation in WT mice under dobutamine-induced stress compared to TRPC6 KO mice.

To control for animal variability within each group, Figure 4 presents changes in HR, EF, FS, SV, CO, LVESV, and LVEDV measured 10 min post-dobutamine injection compared to pre-injection baseline on each animal as indicators of cardiac reserve capacity under dobutamine-induced cardiac stress. No significant HR, EF, or LVESV changes were observed between WT and TRPC6 KO mice, regardless of RD or HFD+L-NAME treatment (Figure 4A,B,E). However, changes in SV and CO were significantly greater in KO HFD+L-NAME mice than in WT HFD+L-NAME mice (Figure 4C,D). Meanwhile, the reduction in LVEDV was considerably less in KO HFD+L-NAME mice than in WT HFD+L-NAME mice under stress conditions (Figure 4F). Representative echocardiographic images for each group are shown in Appendix A. Representative echocardiographic cine loops video comparing between WT HFD+L-NAME and TRPC6 KO HFD+L-NAME mice after dobutamine treatment are shown in Appendix A.

### 2.6. Reduced Exercise Capacity in WT but Not TRPC6 KO Mice After HFD+L-NAME

We conducted a treadmill running test to evaluate exercise capacity in the different groups of mice. There were no significant differences in exercise performance between WT and TRPC6 KO mice on an RD. However, WT mice treated with HFD+L-NAME exhibited significantly reduced exercise tolerance, as evidenced by shorter time and distance to exhaustion (Figure 5A,B) and decreased vertical work after being normalized to body weight (Figure 5C) compared to mice fed with RD. In contrast, exercise tolerance in TRPC6 KO mice on HFD+L-NAME was similar to that of RD-fed mice and significantly higher than that of WT HFD+L-NAME mice.

### 2.7. TRPC6 Gene Expression in Mitochondria and Cytosol of Cardiac Cells in WT and TRPC6 Mice with RD or HFD+L-NAME

We measured TRPC6, TRPC3, PGC-1α, and cleaved caspase 9 and BNP expression levels in the cytosol of cells from the left ventricle of WT and TRPC6 KO mice treated with RD or HFD+L-NAME (Figure 6A). TRPC6 expression levels in the cytosol significantly increased in WT mice after being treated with HFD+L-NAME compared to RD. In contrast, TRPC6 expression levels were very low in TRPC6 KO mice with RD or HFD+L-NAME. For TRPC3, there were no significant differences in its expression levels in WT and TRPC6 KO mice treated with RD or HFD+L-NAME (Figure 6B). We also assessed cleaved caspase 9 expression levels and the cleaved/full-length ratio of caspase 9 as an activation indicator in the mitochondria-mediated apoptotic pathway. We found that cleaved caspase 9 expression levels were significantly elevated in left ventricular samples from WT HFD+L-NAME mice compared to WT RD controls. At the same time, no increases were observed in TRPC6 KO mice treated with HFD+L-NAME (Figure 6B). Additionally, expression levels of PGC1α, a marker of mitochondrial biogenesis, were not significantly different in WT and TRPC6 KO mice on either RD or HFD+L-NAME treatment (Figure 6B). BNP expression levels, as a marker of HFpEF, were measured and showed a significant increase in WT HFD+L-NAME mice compared to WT RD controls and TRPC6 KO treated with HFD+L-NAME (Figure 6B). We further isolated the mitochondrial proteins in left ventricular samples and examined mitochondrial TRPC6 and TRPC3 expression levels in WT and TRPC6 KO mice treated with RD or HFD+L-NAME. Notably, TRPC6 expression levels in mitochondria of WT HFD+L-NAME mice showed a nearly two-fold increase compared to WT RD mice (Figure 6C,D). In contrast, TRPC3 expression remained unchanged in the mitochondria of WT and TRPC6 KO mice, regardless of RD or HFD+L-NAME treatment (Figure 6C,D). Because the TRPC6 signal detected in mitochondrial fractions is low in intensity and subcellular fractionation is inherently semi-quantitative, we interpret these immunoblots as evidence of increased enrichment of TRPC6 immunoreactivity in the mitochondrial fraction after HFD+L-NAME rather than definitive proof of intramitochondrial localization. Densitometry is reported as TRPC6 normalized to VDAC and expressed relative to WT RD.

To confirm Western blot results, we also examined cleaved caspase 9 and BNP expression pattern in the left ventricles of mice using IHC. The results revealed markedly stronger BNP (Figure 7A–H) and cleaved caspase-9 (Figure 7I–P) staining in the endocardial region of WT mice treated with HFD + L-NAME compared to other groups. These findings are consistent with the Western blot results. In WT HFD + L-NAME mice, intense BNP staining indicates heightened myocardial stress, while elevated cleaved caspase-9 reflects activation of mitochondria-initiated apoptotic pathways. In contrast, TRPC6 KO mice under the same conditions exhibited significantly reduced staining, suggesting that TRPC6 deficiency confers protection against metabolic and hypertensive stress-induced cardiac stress and apoptosis.

### 2.8. TRPC6 KO Attenuates Mitochondrial Dysfunction and Reduces Mitochondria-Derived ROS Generation in Mice Treated with HFD+L-NAME

To evaluate the mitochondrial functional changes in WT and TRPC6 KO during the development of HFpEF, we performed detailed mitochondrial respiration measurements in isolated cardiac fibers from the left ventricle. We assessed the maximal oxidative phosphorylation (OXPHOS) capacity induced by complex I and II substrates in the presence of ADP as an indicator of mitochondrial respiration capacity. ATP-linked respiration rate was also measured as an indicator of mitochondrial respiration efficiency by comparing the changes in O_2_ consumption rate after adding the ATP synthase inhibitor oligomycin. The results revealed that WT mice with HFD+L-NAME exhibited significantly reduced maximal and ATP-linked respiration compared with WT mice with RD (Figure 8A,B). Notably, the cardiac fibers isolated from TRPC6 KO mice with HFD+L-NAME showed significantly higher maximal respiration and ATP-linked respiration rates when compared with WT-HFD+L-NAME mice (Figure 8A,B). In addition, we simultaneously measured mitochondrial H_2_O_2_ flux as an indicator of mitochondrial-derived ROS generation at baseline and maximal respiration, and the results indicated a significantly higher H_2_O_2_ flux at baseline and during maximal respiration in the WT mice with HFD+L-NAME compared to WT RD controls. In contrast, mitochondrial H_2_O_2_ flux during maximal respiration was significantly reduced in TRPC6 KO mice with HFD+L-NAME compared with WT mice with HFD+L-NAME (Figure 8C,D).

### 2.9. Cardiac Mitochondria DNA Copy Numbers in Mice with HFD+L-NAME

To determine whether reduced mitochondrial respiration in WT mice after HFD+L-NAME was associated with decreased mitochondrial numbers in the heart, we measured the relative mitochondrial DNA/nuclear DNA ratio as an indicator of mitochondrial number using RT-PCR. The PCR results were normalized using two reference genes: 18S rRNA and lipoprotein lipase (LPL). Our findings revealed no significant changes in mitochondrial DNA copy number in WT or TRPC6 KO mice on RD or HFD+L-NAME when normalized to LPL or 18S rRNA (Figure 8E,F). This finding indicates that mitochondria numbers did not change significantly in both groups of mice after HFD+L-NAME and may not directly contribute to reduced oxygen respiration.

## 3. Discussion

The key findings from our study are as follows: (1) In a model of early-stage HFpEF induced by obesity and mild HTN, we observed that mitochondrial dysfunction, characterized by impaired OXPHOS and increased mitochondria-derived ROS generation, developed before fibrosis. (2) Elevated TRPC6 protein expression was observed in mitochondria and cytosol extracted from left ventricles of WT mice exposed to HFD+L-NAME, suggesting potential roles for mitochondria, as well as membrane-localized TRPC6-mediated Ca^2+^ influx in regulating cellular functions. To our knowledge, this is the first evidence that TRPC6 is localized in cardiac mitochondria. (3) In TRPC6 KO mice, mitochondrial and cardiac diastolic dysfunction are mitigated compared to WT mice during development of HFpEF. These findings indicate that TRPC6 upregulation may play a critical role in the development of HFpEF, potentially by driving mitochondrial dysfunction and activating apoptotic pathways.

We used an HFD+L-NAME mouse model of HFpEF, which has many features associated with HFpEF in humans, including obesity, impaired endothelial function, and increased BP. L-NAME, a NOS inhibitor, reduces NO availability, leading to vasoconstriction [22,23,24] and impaired kidney function, raising arterial BP and inducing hypoxia in cardiac tissue [25,26]. The progression of HFpEF in our model may not be explained solely by the duration of exposure but also by the accumulated metabolic stress from the high-fat diet in combination with the sustained inhibition of nitric oxide synthase by L-NAME. While prolonged exposure likely contributes to the gradual worsening of cardiac function, chronic L-NAME treatment may exert additional effects on vascular regulation, endothelial dysfunction, and possibly central mechanisms of blood pressure control. Future studies will be needed to dissect the relative contributions of these cumulative effects. In the present study, HFD+L-NAME caused an increase in MAP of approximately 17 mmHg, assessed by radio telemetry in conscious WT controls and TRPC6 KO mice. Our preliminary studies found that neither L-NAME nor HFD caused significant diastolic dysfunction. However, the combination of HFD+L-NAME produces mild HTN and metabolic abnormalities that contribute to the development of mitochondrial dysfunction and diastolic dysfunction, which are essential precursors to development of HFpEF. Future studies will be needed to dissect the relative contributions of these cumulative effects versus direct effects.

Our results differ somewhat from Schiattarella et al.’s study [20], which used a 60% HFD combined with L-NAME for 15 weeks in C57BL/6N mice. Although similar pathological phenotypes of HFpEF, including cardiac hypertrophy, were observed in both studies, we did not observe significant cardiac fibrosis after HFD+L-NAME in WT or TRPC6 KO mice. This difference may be due to differences in the length of time for L-NAME treatment, the fat content in the diets, and different strains of mice. However, the primary phenotypes of HFpEF, including cardiac hypertrophy, preserved EF, impaired diastolic function, and reduced exercise tolerance, were similar in both studies.

### 3.1. Diastolic Dysfunction and Impaired Cardiac Reserve in the HFpEF Model 

We observed moderate diastolic dysfunction, represented by a ~40% increase in IVRT and MPI in HFD+L-NAME mice compared with controls. We also performed a dobutamine stress test to evaluate cardiac function when the heart was challenged with increased workload. Dobutamine, a synthetic β-adrenergic agent, increases cardiac contractility and causes mild increases in heart rate and arterial BP. This test is commonly used in human coronary heart disease and HF diagnosis [27,28]. Dobutamine increased HR, SV, CO, EF, and FS in WT mice on RD. Remarkably, WT mice with HFD+L-NAME showed a gradually reduced CO and SV compared to WT control mice during the dobutamine stress test, suggesting impaired cardiac reserve despite preserved HR responses. A probable reason for reduced SV and CO in WT HFD+L-NAME mice during the dobutamine stress test, even when EF and FS were increased, could be the significantly reduced LVEDV, suggesting impaired cardiac filling during diastole. The finding of reduced cardiac reserve is further supported by treadmill exercise tests, which showed significant reductions in running distance, running time, and vertical workload in WT HFD+L-NAME mice compared to WT control mice, indicating ongoing development of cardiac dysfunction.

### 3.2. Mitochondrial Dysfunction Is an Early Event in the Development of HFpEF

We investigated whether mitochondrial dysfunction occurs at the early stage of HFpEF by measuring mitochondrial oxygen consumption rate in cardiac muscle fibers. This approach maintains cell integrity and specifically targets left ventricular cardiac muscle, avoiding interference from other heart cell types [29,30]. Our findings revealed a significant reduction in oxygen consumption rate and increased mitochondrial ROS production in WT HFD+L-NAME mice compared to WT RD controls, indicating impaired OXPHOS. These changes in mitochondrial function suggest that bioenergetic alterations and enhanced mitochondrial ROS production may play pivotal roles in HFpEF pathogenesis. Additionally, the observed mitochondrial dysfunction does not appear to result from reduced mitochondrial numbers since mitochondrial DNA copy numbers and PGC-1α expression levels remained unchanged in WT and TRPC6 KO HFpEF mice.

### 3.3. The Role of TRPC6 in Development of HFpEF

Pharmacological inhibition of TRPC6, a Ca^2+^ channel expressed in the heart, may attenuate HF in the transverse aortic constriction (TAC)-induced pressure overload model by reducing fibrosis and inhibiting cardiac hypertrophy [31,32]. However, TAC induces a rapid onset of pressure overload, unlike the slow, gradual progression of HTN in human HFpEF patients. To our knowledge, our study is the first to identify a cardiac protective role for TRPC6 deficiency in an HFD+L-NAME-induced HFpEF model. We found that TRPC6 KO mice developed cardiac hypertrophy similar to WT mice after HFD+L-NAME treatment, possibly due to HFD-induced obesity and elevated BP. In our model, fibrosis was not significantly detected by Masson’s trichrome staining. This likely reflects the relatively early stage of HFpEF being studied, where diastolic dysfunction, cardiomyocyte hypertrophy, mitochondrial and metabolic alterations are already evident, whereas interstitial and perivascular fibrosis typically emerge at later disease stages. It is also possible that our sample size limited the detection of more subtle fibrotic changes. Future studies with larger cohorts and extended time points will be important to determine whether TRPC6 contributes to the development of myocardial fibrosis as HFpEF progresses. Moreover, TRPC6 KO mice exhibited fibrosis and collagen deposition levels comparable to WT mice after HFD+L-NAME, suggesting that the cardioprotective effects of TRPC6 deficiency in this early stage of HFpEF are independent of fibrosis. Future studies with larger cohorts and extended time points will be essential to determine whether TRPC6 contributes to the development of myocardial fibrosis as HFpEF progresses.

Notably, our study shows increased TRPC6 expression in cardiac mitochondria following HFD+L-NAME treatment, suggesting a potential role for TRPC6 in contributing to enhanced mitochondrial Ca^2+^ influx, which may contribute to mitochondrial Ca^2+^ overload. Chronic Ca^2+^ overload in the mitochondria may disrupt OXPHOS and promote excessive ROS generation, ultimately triggering mitochondrial-induced apoptosis in cardiomyocytes and contributing to the progression of HFpEF [33,34]. However, the differential roles of mitochondrial TRPC6-induced and cell membrane TRPC6 in the development of HFpEF remain unclear and warrant further investigation. The changes observed in TRPC6 KO mice likely arise from altered Ca^2+^-dependent signaling pathways. TRPC6 activates calcineurin–NFAT signaling and contributes to cardiomyocyte hypertrophy and fibroblast-mediated fibrosis. Reductions in TRPC6 activity may therefore blunt these maladaptive responses. In addition, TRPC6 may influence mitochondrial Ca^2+^ homeostasis and ROS generation, linking channel activity to mitochondrial dysfunction in HFpEF. Because TRPC6 is expressed in cardiomyocytes, fibroblasts, and endothelial cells, systemic deletion may affect intercellular communication within the myocardium. These mechanisms together provide a rationale for the phenotypic changes observed in TRPC6 KO mice, though further cell-specific studies will be required to fully define these pathways.

### 3.4. Limitations

One limitation of the present work is the use of global TRPC6 knockout mice. TRPC6 is expressed in multiple tissues, including neurons, podocytes, endothelial cells, fibroblasts, and cardiomyocytes. We previously reported that global TRPC6 deletion alters systemic metabolism, leading to increased body weight and reduced leptin sensitivity [35]. Therefore, we cannot completely exclude systemic influences outside the heart. However, our current findings, together with ongoing studies in cardiac-specific TRPC6 knockout mice, support a direct role for TRPC6 in cardiomyocyte and cardiac remodeling processes in HFpEF. Second, although our Western blotting supports the presence of TRPC6 in cardiac mitochondrial fractions, the relatively lower abundance of TRPC6 in cardiac mitochondria compared to brain mitochondria and the technical limitations of fractionation warrant further validation using complementary methods such as immunofluorescence or high-resolution electron microscopy imaging. Future studies will be directed toward applying these approaches to more definitively establish mitochondrial localization of TRPC6 in the heart.

Another important future direction is to determine whether TRPC6 deficiency influences mitochondrial Ca^2+^ handling through compensatory changes in mitochondrial calcium uniporter (MCU) activity. While the precise interaction between TRPC6 and MCU remains unclear, prior studies have shown that genetic deletion of MCU alone does not confer protection against heart failure [36]. This highlights that mitochondrial Ca^2+^ regulation in heart failure is multifaceted and likely involves additional pathways beyond MCU, warranting further exploration into the broader network of Ca^2+^-handling proteins. An important limitation of the present study is that it uses only male mice. Prior studies have shown that in the C57BL/6 HFpEF model induced by HFD+L-NAME, female mice are relatively protected compared to males, likely due to the cardioprotective effects of estrogen at younger ages. In our ongoing work, we have begun to address this by examining aged cohorts, where we observed HFpEF development in both sexes, suggesting that age-related loss of estrogen’s protective effects may contribute to sex convergence in disease susceptibility. Future studies incorporating sex-based comparisons will be essential to fully define the role of TRPC6 signaling in HFpEF pathogenesis.

## 4. Materials and Methods

### 4.1. Animals

The experimental procedures followed the National Institutes of Health Guide for the Care and Use of Laboratory Animals and had been approved by the Institutional Animal Care and Use Committee of the University of Mississippi Medical Center (Protocol Approval #: 2022-1228). TRPC6 KO mice (B6;129S-TRPC6tm1Lbi/Mmjax) and wild-type (WT) control B6/129s mice, originally from Jackson Laboratories, were bred in our animal facility [35,37,38]. The B6/129s mice were the offspring of C57BL/6J females (B6) and 129S1/SvImJ males (129S) and were crossed for more than four generations and served as controls for TRPC6 KO mice that were generated with 129-derived embryonic stem cells and maintained on a mixed B6/129 background.

### 4.2. Experimental Protocol

Male WT and TRPC6 KO mice were housed in cages maintained at 23 ± 2 °C with a 12:12 h light–dark cycle. We have four groups of mice in this study: (1) WT mice fed a regular diet (WT RD), (2) TRPC6 KO mice fed a regular diet (KO RD), (3) WT mice fed a high-fat diet (HFD; 45% kcal from fat, TD.08811; Inotiv, West Lafayette, IN, USA) and treated with L-NAME (N5751, Sigma-Aldrich, Saint Louis, MO, US) (WT HFD+L-NAME), (4) TRPC6 KO mice fed a high-fat diet and treated with L-NAME (KO HFD+L-NAME).

The protocol to induce HFpEF in groups 3 and 4 mice was modified from a previously reported 2-hit strategy [20]. The mice were fed an HFD for 12 weeks starting at 18 weeks of age. During the first 4 weeks of the HFD, the mice drank tap water, after which L-NAME was added to their drinking water for the remaining 8 weeks (Figure 1). L-NAME (0.5 g/L) was freshly prepared in water every week, and the pH of the L-NAME solution was adjusted to 7.0. Consumption of L-NAME was confirmed by the daily measurement of water intake, which was around 2 mL per day.

Echocardiography was performed in mice of all groups at baseline, at 26 weeks (4 weeks of L-NAME), and 30 weeks of age (8 weeks of L-NAME). A dobutamine stress test and treadmill exercise test were performed at 30 weeks of age, and then hearts were harvested at the end of the experiments for the in vitro studies. Fresh isolated cardiac fibers from left ventricles were used to measure mitochondrial respiration and mitochondrial-derived ROS production (Figure 9). The remaining portions of the heart were stored for morphological analysis and molecular biology protocols.

BP and heart rate (HR) were measured in additional groups of WT and TRPC6 KO mice by radio telemetry at baseline (18 weeks of age) and 30 weeks of age (8 weeks of L-NAME treatment). Briefly, mice at 16 weeks of age were anesthetized with 2% isoflurane, and a telemetry probe (TA11PA-C10, Data Science International, St. Paul, MN, USA) was implanted in the left carotid artery and advanced into the aorta for measurement of BP and HR, 24 h/day [35,37]. The measurements were started after 10 days of recovery from surgery and continued for 5 consecutive days of stable baseline BP and HR recordings. Mean, systolic, and diastolic BP were recorded for 30 s every 10 min, 24 h/day for 5 days at baseline, and then measured again at 30 weeks for 5 days.

### 4.3. Transthoracic Echocardiography

Cardiac function was measured via echocardiography (Vevo 3100, VisualSonics, Toronto, ON, Canada). Animals from each group were anesthetized with 1–2% isoflurane (inhalation), and transthoracic echocardiography was performed to measure cardiac function, including systolic and diastolic parameters. We controlled heart rate at ~400 BPM when measurements were taken. Data from at least three cardiac cycles were collected in each mouse at each time point. Left ventricular trace was performed to obtain an averaged ejection fraction (EF), fractional shortening (FS), stroke volume (SV), cardiac output (CO), and thickness of the left ventricular anterior and posterior wall during systole and diastole (LVAW, s; LVAW, d; LVPW, s; LVPW d) for each group. Diastolic function was assessed by the ratio of early diastolic flow peak velocity of the mitral valve (E) to early diastolic peak velocity of mitral valve annulus (e’) (E/e’), isovolumetric relaxation time (IVRT), isovolumetric contraction time (IVCT), and ejection time (ET) were also obtained by PW Doppler imaging. Myocardial performance index was calculated as (IVCT+IVRT)/ET. The software (Vevo LAB v3.2.0) tracked their movement frame-by-frame throughout the cardiac cycle [39].

### 4.4. Dobutamine Stress Test

At the end of the echocardiography measurements at 30 weeks of age (8 weeks after L-NAME), dobutamine solution (0.1 μg/μL) was injected i.p. at a single 0.75 mg/kg dose. HR, EF, FS, SV, CO, left ventricular end-diastolic volume (LVEDV), and left ventricular end-systolic volume (LVESV) were measured by echocardiography at baseline, 1 min, 5 min, and 10 min post-injection [40,41,42].

### 4.5. Treadmill Exercise Test

A treadmill exercise stress test was performed on WT and TRPC6 KO mice group at 30 weeks of age to examine their cardiac reserve capacity. Mice were acclimated to the treadmill (Columbus Instruments, Columbus, OH, USA) for 2 consecutive days for 5 min at low speed (9 m/min), with no inclination and the shock grid activated. The day after acclimatization, mice were placed on the treadmill at 0° incline, and the shock grid was activated. The treadmill speeds were increased until exhaustion as follows (speed, duration, grade): (9 m/min, 2 min, 5°), (12 m/min, 2 min, 10°), (15 m/min, 2 min, 15°), (18 m/min, 2 min, 15°), (21 m/min, 2 min, 15°), (23 m/min, 2 min, 15°), (24 m/min, 2 min, 15°) and (+1 m/min, each min after that, 15°). Mild shock grid settings were 25 V, 0.34 mA, and 2 Hz. The criteria for exercise-induced exhaustion were (1) 5 consecutive seconds on the electric grid and/or (2) lack of motivation to manual prodding. Each mouse was removed immediately from their respective lane once one or more of these criteria were reached. Following the protocol, the mice were housed separately for 30 min to avoid aggressive interactions following exercise. Running distance was calculated as speed X running time, while vertical workload was calculated as body weight X vertical distance [43].

### 4.6. High-Resolution Respirometry (HRR) Measurement of Mitochondrial Respiration in Permeabilized Heart Samples

Mitochondrial respiratory rate and mitochondrial-derived ROS production were measured simultaneously in Saponin-permeabilized cardiac fibers by high-resolution respirometry (Oroboros Instruments—Oxygraph-2k, Innsbruck, Austria), according to substrate–uncoupler–inhibitor–titration (SUIT) protocol as previously described [44,45]. Briefly, thin cardiac fibers from left ventricle (around 1.5–2.0 mg per sample) were incubated with 100 μg/mL Saponin dissolved in relaxation and preservation solution (7.2 mM K_2_EGTA, 2.8 mM CaK_2_EGTA, 20 mM Imidazole, 0.5 M Dithiothreitol, 20 mM Taurine, 5.7 mM ATP, 14.3 mM Phosphocreatine, 6.6 mM MgCl_2_, 50 mM K-MES, pH 7.1) on a shaker at 50 rpm for 40 min, followed by 3x washes with MiRO5 buffer (60 mM K-lactobionate, 0.5 mM EGTA, 3 mM MgCl_2_, 20 mM Taurine, 10 mM KH_2_PO4, 20 mM HEPES, 110 mM sucrose, and 1 g/L BSA, pH 7.1) for 10 min each. Cardiac fibers were weighed and transferred to the Oxygraph-2k chambers with MiRO5 solution at 37 °C [29]. Oxygen was added to the solution in each chamber to increase oxygen concentration above 350 μM. After oxygen concentration stabilized, mitochondrial respiration rate was measured as follows: addition of glutamate (final concentration, 10 mM) and malate (2 mM) as complex I substrates; addition of ADP (5 mM) to activate oxidative phosphorylation (OXPHOS), succinate (20 mM), a complex II substrate addition to further stimulate respiration by activating convergent electron flow from complex I+II into the Q junction, a step referred as maximum OXPHOS capacity; addition of ATP synthase inhibitor, oligomycin (5 μM), to measure the ATP-linked respiration rate. Inhibition of respiration with complex I inhibitor rotenone (1 μM) and complex III inhibitor antimycin A (5 μM) was added to measure non-OXPHOS respiration. Substrate and inhibitor concentrations were selected based on previous experimental determination of doses needed to stimulate or reduce maximal respiration rate. Finally, data analysis was performed using DATLAB 4.2 software (Oroboros Instruments), and tissue respiration was expressed as oxygen flux (pmol/s/mg).

### 4.7. Hydrogen Peroxide Flux Measurement as an Indicator of Mitochondrial Superoxide Generation in Permeabilized Heart Samples

Mitochondrial superoxide flux was measured simultaneously with respirometry in the O2k-Fluorometer using the H_2_O_2_-sensitive probe Amplex UltraRed (Invitrogen, Carlsbad, CA, USA). Amplex UltraRed (AmR, 10 µM), 1 U/mL horse radish peroxidase (HRP), and 5 U/mL superoxide dismutase were added to the MiRO5 solution. The reaction product between AmR and H_2_O_2_, catalyzed by HRP, is fluorescent. Volume-specific H_2_O_2_ fluxes were calculated in real time by the DatLab software (Oroboros Instruments, Innsbruck, Austria) from the time derivative of the fluorescent signal over time [46].

### 4.8. Fractional Mitochondria and Cytosol Protein Isolation for Western Blot

Cardiac mitochondria were isolated from the left ventricles of WT and TRPC6 KO mice using a mitochondria isolation kit (Novus Biologicals, Littleton, CO, USA, NBP2-29448). The cytosol fraction was obtained in the supernatant, and the mitochondrial pellet was lysed using a mitochondrial lysis buffer. After protein concentration was measured by BCA assay, a total of 15 µg of protein from cytosol and mitochondria was separated in 7.5% SDS-polyacrylamide gels (Bio-rad, Hercules, CA, USA). After transfer to nitrocellulose membranes, blots were rinsed in PBS and blocked in Odyssey blocking buffer (LI-CORbio, Lincoln, NE, USA) for 1 h at room temperature and then incubated with TRPC6 rabbit antibody (1:500, ACC-120, Alomone Labs, Jerusalem, Israel) or TRPC3 rabbit antibody (1:500, #77934, Cell Signaling Technology, Danvers, MA, USA), VDAC rabbit antibody (1:1000, #4661, Cell Signaling), GAPDH rabbit antibody (1:2000, #2118, Cell Signaling), caspase 9 rabbit antibody (1:1000, #9504, Cell Signaling), cleaved caspase 9 rabbit antibody (1:1000, #7237, Cell Signaling), PGC-1α rabbit antibody (1:1000, #2178, Cell Signaling) and BNP rabbit antibody (1:1000, PA5-96084, ThermoFisher, Waltham, MA, USA) at 4 °C overnight. Membranes were probed with LI-COR fluorescent dye-labeled secondary antibodies (1:5000) for 1 h at room temperature. Antibody labeling was visualized using the Odyssey Infrared Scanner (LI-CORbio). GPADH was used as the internal loading control for the cytosolic fraction, and VDAC was used as the internal loading control for the mitochondrial fraction. Polyacrylamide gels ranging from 7.5% to 12% were used: lower-percentage gels (7.5–8%) were used for higher molecular weight proteins, whereas higher-percentage gels (10–12%) were used for lower molecular weight proteins.

### 4.9. Immunohistochemistry (IHC)

Left ventricular tissues were harvested, fixed in 4% paraformaldehyde, embedded in paraffin, and sectioned at 5 µm thickness. After deparaffinization and rehydration, sections were subjected to antigen retrieval using citrate buffer (pH 6.0) followed by blocking endogenous peroxidase activity with 3% hydrogen peroxide. Non-specific binding was blocked with 5% normal goat serum, and sections were incubated overnight at 4 °C with primary antibodies against BNP (1:100, PA5-96084, ThermoFisher, Waltham, MA, USA) and cleaved caspase-9 (1:200, #7237, Cell Signaling). After washing, sections were incubated with HRP-conjugated secondary antibodies, developed with DAB substrate, and counterstained with hematoxylin. Images were acquired using a light microscope, and staining was evaluated in the left ventricular region.

### 4.10. Evaluation of Cardiac Fibrosis

Heart paraffin-embedded tissue sections (5 μM) were prepared as described previously [47]. Masson trichrome stain (Sigma-Aldrich, HT15-1KT) was used to detect fibrosis [39]. The cytoplasm and muscle fibers are stained red, while collagen fibers are stained blue. Images showing regions of interest (ROI) were analyzed using ImageJ 1.54 software to measure the blue versus red area [48].

### 4.11. Mitochondrial DNA Content Measurement by Real-Time Quantitative PCR

Total mitochondrial DNA was isolated from left ventricle of the heart tissues using Trizol reagent (Invitrogen, 15596026) and real-time qPCR was performed as previously described [35]. PCR was performed in a 10 μL reaction mixture prepared with SYBR GREEN PCR Master Mix (Applied Biosystems, Warrington, UK) containing an appropriately diluted cDNA solution and 0.2 μM of each primer at 95 °C for 10 min, followed by 35 cycles at 95 °C for 10 s and 60 °C for 45 s. All reactions were conducted in triplicate, and data were averaged and analyzed using the delta Ct (∆∆Ct) method. Mitochondrial gene COX3 was measured and normalized to chromosome lipoprotein lipase gene and 18s rRNA gene. The details of primers are COX3: forward AGCCCATGACCACTAACAGG; reverse CGTGGGTAGGAACTAGGCTG, 18s rRNA: forward GCGGTTCTATTTTGTTGGTTTT; reverse ACCTCCGACTTTCGTTCTTG, lipoprotein lipase: forward GGATGGACGGTAAGAGTGATTC; reverse ATCCAAGGGTAGCAGACAGGT.

### 4.12. Data Representation and Statistical Analysis

Data were presented as mean ± standard error of the mean (SEM). For animal groups with repeated measurements, two-way ANOVA followed by Tukey’s multiple comparisons test was applied. For comparisons among multiple groups, one-way ANOVA followed by Tukey’s post hoc test was performed. The statistical methods applied for each comparison are detailed in the figure legends. Statistical analyses were conducted using Prism (version 9.3.1; GraphPad Software, La Jolla, CA, USA), and a *p*-value of less than 0.05 was considered statistically significant.

## 5. Conclusions

To our knowledge, our study is the first to demonstrate TRPC6 expression in mitochondria and show that mitochondrial and cytosolic TRPC6 expression are increased in cardiac tissue of mice with HFpEF induced by HFD+L-NAME. Increased TRPC6 expression was associated with elevated mitochondria-derived ROS generation and mitochondria-mediated apoptosis pathway activation. We also found that TRPC6 deficiency effectively attenuates diastolic dysfunction while preserving cardiac reserve capacity in the HFD+L-NAME HFpEF model. Additionally, TRPC6 deficiency markedly attenuated mitochondrial dysfunction and decreased mitochondrial ROS generation in this model. Importantly, our findings reveal a previously unrecognized role of TRPC6 as a direct regulator of mitochondrial homeostasis in the failing heart. By linking TRPC6 activity to both cytosolic signaling and mitochondrial function, this study identifies TRPC6 as a dual-compartment modulator of cardiac pathophysiology. This dual role not only expands the current understanding of ion channel biology in HFpEF but also highlights TRPC6 as a uniquely attractive therapeutic target. While our results do not establish TRPC6 deficiency as a therapeutic strategy, they provide new mechanistic insight and suggest that modulating TRPC6 activity could represent a potential avenue for therapeutic intervention in HFpEF.

## Figures and Tables

**Figure 1 ijms-26-09383-f001:**
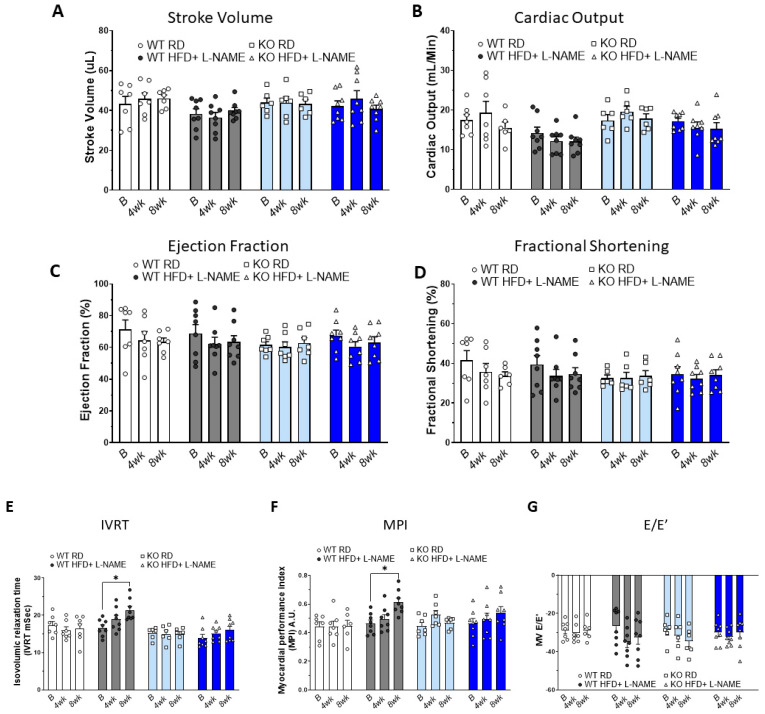
**Cardiac systolic and diastolic functions assessed by echocardiography in WT and TRPC6 KO.** (**A**) Stroke volume, (**B**) cardiac output, (**C**) ejection fraction, (**D**) fractional shortening, (**E**) isovolumic relaxation time, (**F**) myocardial performance index, and (**G**) E/e’, measurements at baseline, 4 weeks, and 8 weeks after L-NAME. Results are expressed as mean ± SEM, *n* = 6–7, * *p* < 0.05 between two groups by post hoc test after two-way ANOVA. Bar colors indicate experimental groups: white = WT RD (regular diet), gray = WT HFD+L-NAME, light blue = KO RD, dark blue = KO HFD+L-NAME.

**Figure 2 ijms-26-09383-f002:**
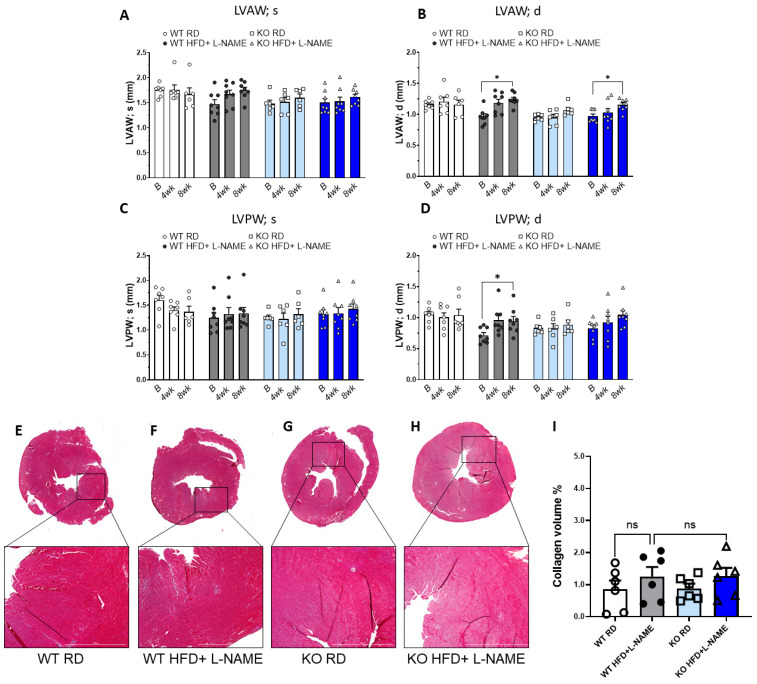
**Left ventricle wall thickness and fibrosis in WT and TRPC6 KO mice.** (**A**) End-systolic left ventricular anterior wall thickness, (**B**) end-diastolic left ventricular anterior wall thickness, (**C**) end-systolic left ventricular posterior wall thickness, and (**D**) end-diastolic left ventricular posterior wall thickness, with measurements at baseline, 4 weeks and 8 weeks after L-NAME. Results are expressed as mean ± SEM, *n* = 6–8, * *p* < 0.05 between two groups by post hoc test after two-way ANOVA. Myocardial fibrosis measurements in WT and TRPC6 KO mice. Representative Masson Trichrome staining of left ventricles from WT RD (**E**), WT HFD+ L-NAME (**F**), KO RD (**G**), and KO HFD+ L-NAME (**H**). (**I**) Quantitative analysis of collagen fractions (blue color) by Masson Trichrome staining in the left ventricles. Results are expressed as mean ± SEM, *n* = 6, * *p* < 0.05 between two groups by non-parametric test after one-way ANOVA. ns, not significant. Bar colors indicate experimental groups: white = WT RD (regular diet), gray = WT HFD+L-NAME, light blue = KO RD, dark blue = KO HFD+L-NAME.

**Figure 3 ijms-26-09383-f003:**
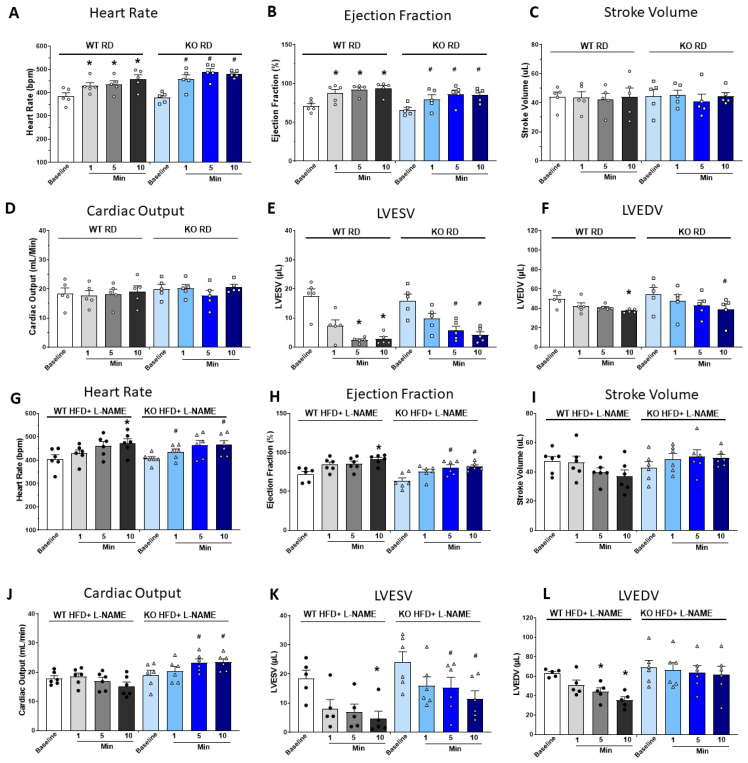
**Echocardiography during dobutamine stress test.** Time-dependent changes after dobutamine injection in (**A**) Heart rate, (**B**) ejection fraction, (**C**) stroke volume, (**D**) cardiac output, (**E**) LVESV, and (**F**) LVEDV in WT and TRPC6 KO mice with RD. (**G**–**L**) Heart rate, ejection fraction, stroke volume, cardiac output, LVESV, and LVEDV in WT and TRPC6 KO mice with HFD+L-NAME after dobutamine injection. Results are expressed as mean ± SEM, *n* = 5–6, *, *p* < 0.05 when compared with baseline in WT mice, #, *p* < 0.05 when compared to baseline in TRPC6 KO mice by post hoc test after one-way ANOVA. LVESV, left ventricular end-systolic volume; LVEDV, left ventricular end-diastolic volume. Color code: for WT RD and WT HFD+L-NAME mice, white (baseline), light gray (1 min), medium gray (5 min), dark gray (10 min) after dobutamine injection; for KO-RD and KO HFD+L-NAME mice, light blue (baseline), medium blue (1 min), darker blue (5 min), dark blue (10 min) after dobutamine injection.

**Figure 4 ijms-26-09383-f004:**
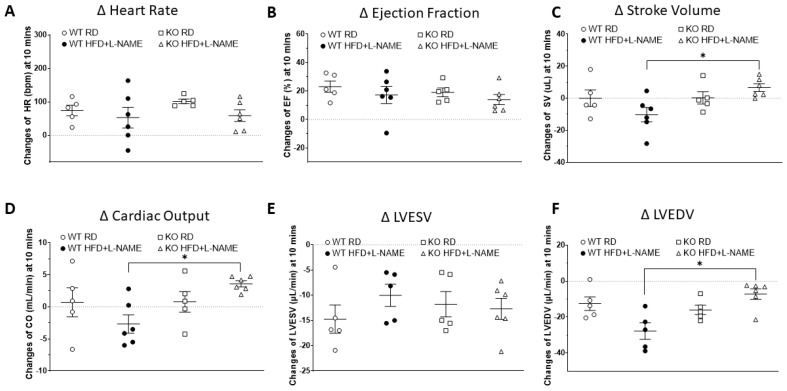
**Changes in cardiac function at 10 min after dobutamine compared to baseline.** Changes in (**A**) heart rate, (**B**) ejection fraction, (**C**) stroke volume, (**D**) cardiac output, (**E**) LVESV, and (**F**) LVEDV in WT and TRPC6 KO mice with RD or HFD+ LNAME. Results are expressed as mean ± SEM, *n* = 5–6, * *p* < 0.05 between two groups by post hoc test after one-way ANOVA.

**Figure 5 ijms-26-09383-f005:**
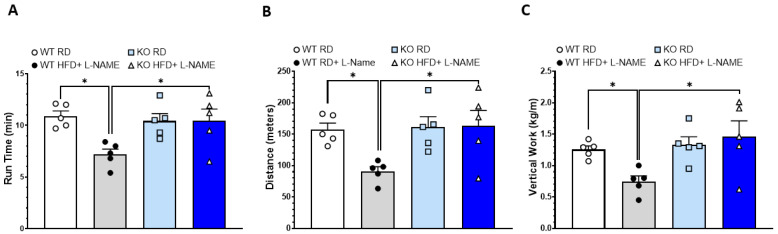
**Treadmill running test in WT and TRPC6 KO mice.** (**A**) Running time, (**B**) running distance, and (**C**) vertical work in WT and TRPC6 KO mice with RD or HFD+ L-NAME. Results are expressed as mean ± SEM, *n* = 5, * *p* < 0.05 between two groups by post hoc test after one-way ANOVA. Bar colors indicate experimental groups: white = WT RD (regular diet), gray = WT HFD+L-NAME, light blue = KO RD, dark blue = KO HFD+L-NAME.

**Figure 6 ijms-26-09383-f006:**
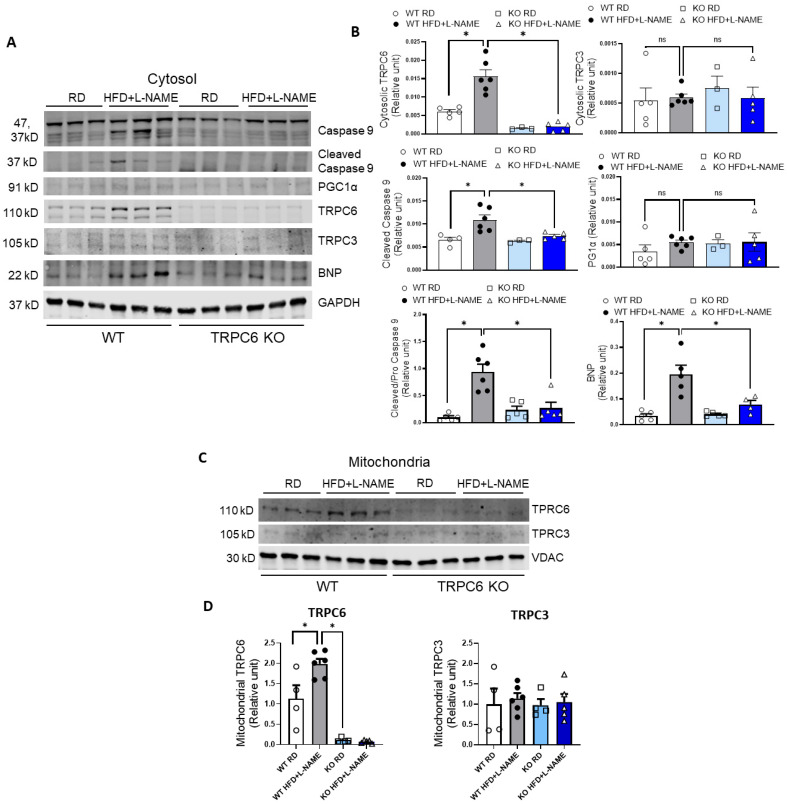
Changes in protein expression within the cytosol and mitochondria isolated from the left ventricle of WT and TRPC6 KO mice. (**A**) Representative blots for caspase 9, cleaved caspase 9, PGC1a, TRPC6, TRPC3, BNP, and GAPDH in the cytosol fraction. (**B**) Quantitative analysis results for protein expression levels in the cytosol. (**C**) Representative blots for TRPC6, TRPC3, and VDAC in mitochondria isolation. (**D**) Quantitative analysis results for protein expression levels in mitochondria. Results are expressed as mean ± SEM, *n* = 3–6, * *p* < 0.05 between two groups by post hoc test after one-way ANOVA. Bar colors indicate experimental groups: white = WT RD (regular diet), gray = WT HFD+L-NAME, light blue = KO RD, dark blue = KO HFD+L-NAME.

**Figure 7 ijms-26-09383-f007:**
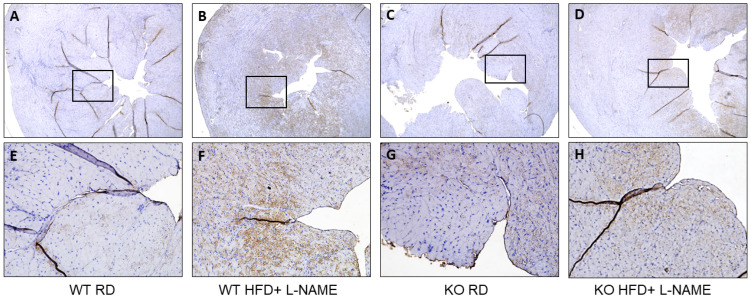
Immunohistochemistry staining of BNP and cleaved caspase 9 from the left ventricle of WT and TRPC6 KO mice with RD or HFD+ L-NAME. Representative images for BNP staining in WT RD (**A**,**E**), WT HFD+ L-NAME (**B**,**F**), KO RD (**C**,**G**), and KO HFD+ L-NAME (**D**,**H**), and cleaved caspase 9 staining in WT RD (**I**,**M**), WT HFD+ L-NAME (**J**,**N**), KO RD (**K**,**O**), and KO HFD+ L-NAME (**L**,**P**). Upper panel, ×40 magnification; lower panel, boxed regions in ×200 magnification.

**Figure 8 ijms-26-09383-f008:**
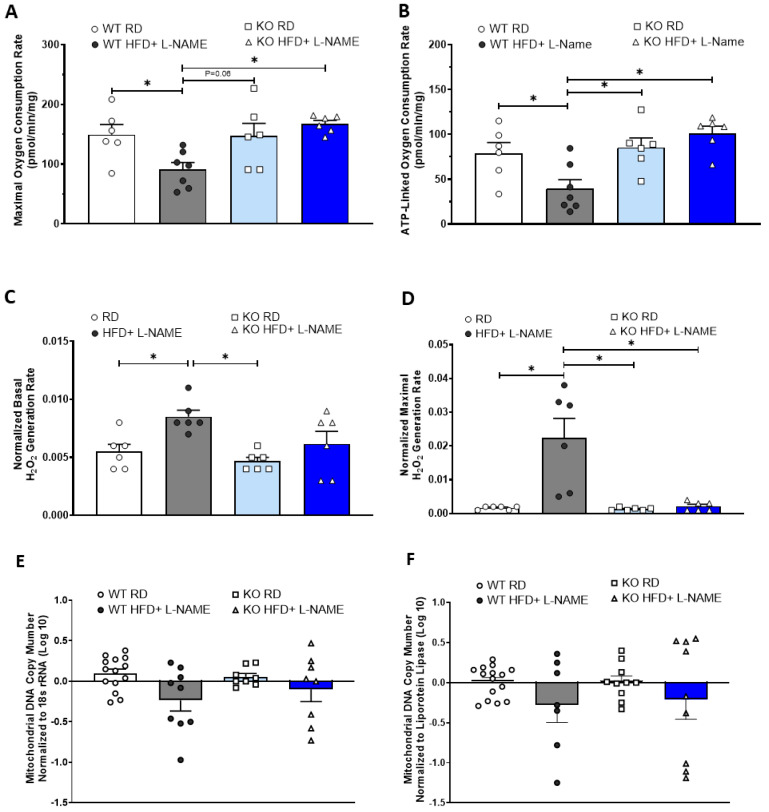
Mitochondrial respiration, mitochondria-derived H_2_O_2_ production, and mitochondrial DNA copy numbers in WT and TRPC6 KO mice. (**A**) Maximal oxygen consumption rate, (**B**) ATP-linked oxygen consumption rate, (**C**) basal H_2_O_2_ generation rate, and (**D**) maximal H_2_O_2_ generation rate measurements by Oroboros respirometer in WT and TRPC6 KO mice with RD or HFD+ L-NAME. (**E**,**F**) Mitochondrial DNA copy number normalized to 18s rRNA and lipoprotein lipase, respectively. Results are expressed as mean ± SEM, *n* = 6, * *p* < 0.05 between two groups by post hoc test after one-way ANOVA. Bar colors indicate experimental groups: white = WT RD (regular diet), gray = WT HFD+L-NAME, light blue = KO RD, dark blue = KO HFD+L-NAME.

**Figure 9 ijms-26-09383-f009:**
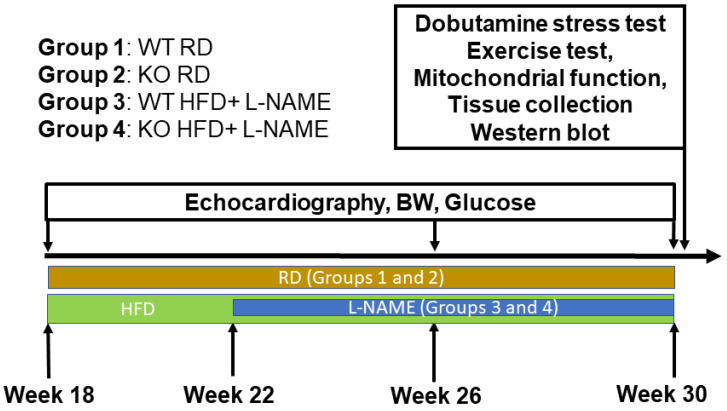
Scheme of the animal groups and protocol.

**Table 1 ijms-26-09383-t001:** Basic physiological parameters in WT and TRPC6 KO mice at the beginning and end of the protocol.

	WT RD		WT HFD+ L-NAME		TRPC6 KO RD		TRPC6 KO HFD+ L-NAME	
	Baseline	30 wks	Baseline	30 wks	Baseline	30 wks	Baseline	30 wks
Body Weight (g)	33.7 ± 4.8	37.7 ± 2.1	35.1 ± 5.6	48.1 ± 3.1 *	37.5 ± 2.7	40.1 ± 1.8	39.7 ± 3.0	45.8 ± 4.0 *
Blood Glucose (mg/dL)	121 ± 6	130 ± 8	127 ± 11	142 ± 9 *	134 ± 8	138 ± 10	142 ± 7	155 ± 10 *
Heart Weight (mg)	N/A	179 ± 5	N/A	237 ± 9 ^+^	N/A	173± 7	N/A	235 ± 11 ^+^
Mean arterial pressure (mmHg)	N/A	N/A	102 ± 3	119 ± 4 *	N/A	N/A	106 ± 8	120 ± 8 *
Heart Rate (beat/min)	N/A	N/A	541 ± 23	515 ± 18	N/A	N/A	578 ± 22	562 ± 28

(*n* = 6–7 per group, *, *p* < 0.05 when compared to baseline, ^+^, *p* < 0.05 when compared to RD).

## Data Availability

The datasets generated and analyzed during the current study are included in this published article and its Appendix A.

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
