# Peer review of "TRPC6 Deficiency Attenuates Mitochondrial and Cardiac Dysfunction in Heart Failure with Preserved Ejection Fraction Induced by High-Fat Diet Plus L-NAME"

_ijms, 2025, doi:10.3390/ijms26199383_

Round 1

Reviewer 1 Report

Comments and Suggestions for Authors

This study is conducted with integrity and transparency which is appreciated. Raw blots and single data points are appreciated. I provide major and minor comments below:

Major

Authors should show representative images of LV 4 wall for cardiac function as well as LV mass data as obtained via echo. Further, authors should show representative images for diastolic function. LV mass via 4 wall measurement is a less subjective measure of LV wall thickness compared to LVAW and LVPW. It can be easily obtained during the LV 4 wall measurement in the Vevo system.

figure 2: XY stitch of entire tissue section should be shown. As it stands, these images are unconvincing, and the stats do not reflect the images. Note that blood vessels will appear as small circular regions surrounded by collagen. These areas should not be mistaken for fibrosis.

Figure 3:  This is not a measure of coronary flow reserve. I direct authors to the following jove protocol. Authors should correct the framing of this data.
https://pmc.ncbi.nlm.nih.gov/articles/PMC6996202/

Was a target heart rate maintained under anesthesia with Echo? Stroke volume and cardiac output are functions of heart rate. Thus, if animals are anesthetized lightly or deeply (if this fluctuates), this can impact heart rate and thus impact stroke volume and cardiac output measures.

Figure 6: Considering the low quality of cleaved caspase 9, authors must confirm the MW as accurate with a positive and negative control lysate or otherwise. This may as well be nonspecific binding otherwise.

Minor

Please provide institutional approval number

The gel % is stated as 7.5%. Are authors sure about this percentage? Low MW proteins are not going to be seen with this percent of acrylamide.

Catalog numbers of antibodies should be provided.

Author Response

Reviewer #1:

Comment 1: Authors should show representative images of the LV 4 wall for cardiac function, as well as LV mass data as obtained via echo. Further, authors should show representative images for diastolic function. LV mass via 4 wall measurement is a less subjective measure of LV wall thickness compared to LVAW and LVPW. It can be easily obtained during the LV 4 wall measurement in the Vevo system.

Response: We added representative echocardiographic images as suggested in supplemental Figures S2-4.

Comment 2: figure 2: XY stitch of entire tissue section should be shown. As it stands, these images are unconvincing, and the stats do not reflect the images. Note that blood vessels will appear as small circular regions surrounded by collagen. These areas should not be mistaken for fibrosis.

Response: As requested, we provided stitched images of the full tissue sections (Figure 3 E-H), allowing for a more comprehensive visualization of the histological context. Additionally, we carefully re-evaluated fibrosis quantification, ensuring that blood vessels were not misinterpreted as fibrotic areas. The fibrosis quantification has been updated accordingly (Figure 3 I), and the revised statistical analysis now more accurately reflects the histological findings.

Comment 3: Figure 3:  This is not a measure of coronary flow reserve. I direct authors to the following jove protocol. Authors should correct the framing of this data.
https://pmc.ncbi.nlm.nih.gov/articles/PMC6996202/

Response: We thank the reviewer for highlighting this important point and directing us to the referenced JoVE protocol. Despite several attempts to measure coronary flow in several mice, we were unable to obtain consistent results, most likely due to the overweight phenotype of the animals in this study. Consequently, we were unable to provide a reliable assessment of coronary flow reserve in mice at this stage.

Comment 4: Was a target heart rate maintained under anesthesia with Echo? Stroke volume and cardiac output are functions of heart rate. Thus, if animals are anesthetized lightly or deeply (if this fluctuates), this can impact heart rate and thus impact stroke volume and cardiac output measures.

Response: During anesthesia and echocardiography, we controlled heart rate at ~400 BPM when measurements were taken. We added this information to our methods on page 5. 

Comment 5: Figure 6: Considering the low quality of cleaved caspase 9, authors must confirm the MW as accurate with a positive and negative control lysate or otherwise. This may as well be nonspecific binding otherwise.

Response: To address the concern regarding the specificity and accuracy of cleaved caspase-9 detection, we performed two additional validation experiments:

  1. Alternative antibody validation: We repeated the western blot using a different caspase-9 antibody that recognizes both the full-length and cleaved forms of caspase-9. Consistent with our original findings, this antibody also detected an increased cleaved caspase-9 level. This result is now shown in our revised Figure 7, confirming the specificity of the band detected.
  2. Immunohistochemistry confirmation: We further examined cleaved caspase-9 expression in the left ventricle using IHC. The observed changes in cleaved caspase-9 expression were consistent with the western blot findings. These results are now presented in Figure 8 of the revised manuscript.

Together, these additional experiments strengthen the evidence that the detected band corresponds specifically to cleaved caspase-9 rather than nonspecific binding.

Comment 6: Please provide institutional approval number

Response: We added the institutional approval number in the methods on page 4.

Comment 7: The gel % is stated as 7.5%. Are authors sure about this percentage? Low MW proteins are not going to be seen with this percent of acrylamide.

Response: We used different percentages of SDS-PAGE gels depending on the molecular weight of the target proteins. Specifically, gels ranged from 7.5% to 12% acrylamide: lower-percentage gels (7.5–8%) were used for higher molecular weight proteins, whereas higher-percentage gels (10–12%) were used to resolve lower molecular weight proteins. We revised the methods section to clarify this point on page 7.

Comment 8: Catalog numbers of antibodies should be provided.

Response: We added catalog numbers to each antibody in the method section on page 7.

Reviewer 2 Report

Comments and Suggestions for Authors

In the article titled "TRPC6 deficiency attenuates mitochondrial and cardiac dysfunction in heart failure with preserved ejection fraction induced by high-fat diet plus L-NAME", the authors explore whether TRPC6 deficiency prevents mitochondrial dysfunction and offers cardiac protection in a mouse model of HFpEF induced by high-fat diet (HFD) for 12 weeks combined with L-NAME administration during the final 8 weeks in TRPC6 knockout (KO) and wild-type (WT) control mice. They conclude that TRPC6 deficiency protects against the development of HFpEF by mitigating diastolic dysfunction, preserving cardiac reserve capacity, and attenuating mitochondrial dysfunction.

The article is very well structured and has a high degree of novelty.

Major concerns:

  1. Did you only use diastolic dysfunction to define HFpEF? According to the ESC guidelines for HF from 2021, the definition of HFpEF also includes atrial natriuretic peptides, which objectively characterizes the occurrence of heart failure.
  2. To make the article much more attractive, please insert echocardiographic images with the determination of LVEF, diastolic dysfunction and other parameters analyzed in your research.

Minor Concerns

  1. Please clearly state the sex of the animals. Only Male WT is specified, but the sex of the other groups is not specified. This could influence the results, as you stated in the limitations.

Author Response

Reviewer #2:

Comment 1: Did you only use diastolic dysfunction to define HFpEF? According to the ESC guidelines for HF from 2021, the definition of HFpEF also includes atrial natriuretic peptides, which objectively characterizes the occurrence of heart failure.

Response: In our initial submission, we defined HFpEF primarily based on evidence of diastolic dysfunction. In this revision, we expanded our analysis to include BNP as an objective marker of heart failure. While ANP is a natriuretic peptide reflecting atrial stretch, BNP is much more reflective of ventricular dysfunction. Therefore, we assessed BNP levels as a representative biomarker consistent with guideline recommendations. Our new results section showed BNP expression by western blot and immunohistochemistry in left ventricular tissue. Both approaches demonstrated consistent changes in BNP levels that aligned with our diastolic dysfunction findings. These new results are presented in Figure 8 of the revised manuscript.

Comment 2: To make the article much more attractive, please insert echocardiographic images with the determination of LVEF, diastolic dysfunction and other parameters analyzed in your research.

Response: We added representative echocardiographic images in supplemental Figures S2-4.

Comment 3: Please clearly state the sex of the animals. Only Male WT is specified, but the sex of the other groups is not specified. This could influence the results, as you stated in the limitations.

Response: All mice in our study were male. We clarified the sex in Methods and Materials. We also discussed the limitations of using only 'male' in the limitations in the discussion section as follows in page 23:

“An important limitation of the present study is that it uses only male mice. Prior studies have shown that in the C57BL/6 HFpEF model induced by HFD+L-NAME, female mice are relatively protected compared to males, likely due to the cardioprotective effects of estrogen at younger ages. In our ongoing work, we have begun to address this by examining aged cohorts, where we observed HFpEF development in both sexes, suggesting that age-related loss of estrogen’s protective effects may contribute to sex convergence in disease susceptibility. Future studies incorporating sex-based comparisons will be essential to fully define the role of TRPC6 signaling in HFpEF pathogenesis”.

Reviewer 3 Report

Comments and Suggestions for Authors

The manuscript entitled "TRPC6 deficiency attenuates mitochondrial and cardiac dysfunction in heart failure with preserved ejection fraction induced by high-fat diet plus L-NAME" investigates the role of Transient Receptor Potential Cation Channel 6 (TRPC6) in cardiac dysfunction, with a focus on heart failure with preserved ejection fraction (HFpEF). The authors illustrate that TRPC6 expression is elevated in both mitochondrial and cytosolic compartments of cardiac tissue in mice presenting with HFpEF as a result of high-fat diet and L-NAME treatment. This heightened expression correlates with increased generation of reactive oxygen species (ROS) and the activation of apoptotic pathways within the mitochondria. In conclusion, the study offers novel insights into the function of TRPC6 in cardiac mitochondria and its potential implications for elucidating the mechanisms underlying HFpEF, suggesting that targeting TRPC6 may provide therapeutic strategies for the management of this condition. The authors are encouraged to provide a comprehensive response to the concerns raised below.

Concerns:

  1. While the manuscript presents evidence regarding TRPC6's role in mitochondrial dysfunction, further examination of its specific mechanisms, such as its interaction with mitochondrial calcium transporters, could yield deeper insights.
  2. Only male mice were utilized, which is concerning given that HFpEF has a higher prevalence in females. The authors should incorporate sex-based comparisons in future studies and briefly discuss how sex differences may influence TRPC6 signaling in cardiac tissue.
  3. The authors acknowledge that the utilization of global TRPC6 knockout (KO) mice is a limitation. To substantiate cardiomyocyte-autonomous effects, it is suggested that they perform or reference supporting data from cardiomyocyte-specific TRPC6 knockouts or isolated cardiomyocytes.
  4. The abstract and discussion sections underscore TRPC6 as a potential therapeutic target; however, the findings' novelty can be further emphasized.
  5. Compensatory mechanisms resulting from the global KO, particularly from other TRPC channels such as TRPC3, may exist. The authors are encouraged to provide additional data, even in supplementary material, regarding the upregulation of other TRPC channels, particularly in the heart or mitochondria. This should be noted as a limitation or an avenue for future research.
  6. Throughout the manuscript, ultrasound-based echocardiography was employed to illustrate the HEpEF; however, the authors did not incorporate specific cine loops. It is advised that the authors include cine loops for each result.
  7. The authors are encouraged to include data that represent hemodynamic changes, specifically systolic and diastolic blood pressure.

Author Response

Reviewer #3:

Comment 1: While the manuscript presents evidence regarding TRPC6's role in mitochondrial dysfunction, further examination of its specific mechanisms, such as its interaction with mitochondrial calcium transporters, could yield deeper insights.

Response: We thank the reviewer for this insightful suggestion. We fully agree that assessing mitochondrial calcium handling would provide a more direct link between TRPC6 expression changes and mitochondrial dysfunction. We are actively investigating mitochondrial calcium dynamics in our model, and we included this as an important future direction in the revised Discussion section.

Regarding the potential interaction between TRPC6 and the mitochondrial calcium uniporter (MCU), this relationship remains unclear. It is not known whether TRPC6 deficiency could lead to compensatory changes in MCU activity. Interestingly, previous studies have shown that genetic deletion of MCU did not protect against the development of heart failure (J Mol Cell Cardiol 85 (2015) 178–182). This suggests that the regulation of mitochondrial calcium homeostasis in heart failure is likely more complex and involves additional mechanisms beyond MCU alone. We included this as an important future direction in the revised Discussion section as follows in page 23:

Another important future direction is to determine whether TRPC6 deficiency influences mitochondrial calcium handling through compensatory changes in MCU activity. While the precise interaction between TRPC6 and MCU remains unclear, prior studies have shown that genetic deletion of MCU alone does not confer protection against heart failure (J Mol Cell Cardiol 85 (2015) 178–182; Cell Rep 12(1) (2015) 15–22). This highlights that mitochondrial calcium regulation in heart failure is multifaceted and likely involves additional pathways beyond MCU, warranting further exploration into the broader network of calcium-handling proteins.

Comment 2: Only male mice were utilized, which is concerning given that HFpEF has a higher prevalence in females. The authors should incorporate sex-based comparisons in future studies and briefly discuss how sex differences may influence TRPC6 signaling in cardiac tissue.

Response: We agree that using only male mice is a limitation of the present study. Indeed, several studies have reported that in the C57BL/6 HFpEF model induced by HFD+L-NAME, female mice are relatively protected compared to males. This protection has been attributed, at least in part, to the cardioprotective effects of estrogen at younger ages.

In our ongoing work, we have begun to investigate sex differences in aged mice. Interestingly, in older male and female mice, we observed the development of HFpEF in both sexes, suggesting that age-related loss of estrogen’s protective effects may contribute to sex convergence in disease susceptibility.

We will add these points as below to the revised Discussion section in page 23, and we acknowledge that incorporating sex-based comparisons is an important future direction for elucidating the role of TRPC6 signaling in HFpEF.

An important limitation of the present study is that it uses only male mice. Prior studies have shown that in the C57BL/6 HFpEF model induced by HFD+L-NAME, female mice are relatively protected compared to males, likely due to the cardioprotective effects of estrogen at younger ages. In our ongoing work, we have begun to address this by examining aged cohorts, where we observed HFpEF development in both sexes, suggesting that age-related loss of estrogen’s protective effects may contribute to sex convergence in disease susceptibility. Future studies incorporating sex-based comparisons will be essential to fully define the role of TRPC6 signaling in HFpEF pathogenesis”.

Comment 3: The authors acknowledge that the utilization of global TRPC6 knockout (KO) mice is a limitation. To substantiate cardiomyocyte-autonomous effects, it is suggested that they perform or reference supporting data from cardiomyocyte-specific TRPC6 knockouts or isolated cardiomyocytes.

Response: We acknowledge in the discussion that the use of global TRPC6 knockout (KO) mice is a limitation of the present study. In future studies of HFpEF, we plan to develop and utilize cardiomyocyte-specific TRPC6cKO mice to more precisely define its contribution to disease pathogenesis.

Comment 4: The abstract and discussion sections underscore TRPC6 as a potential therapeutic target; however, the findings' novelty can be further emphasized.

Response: We thank the reviewer for this constructive suggestion. In the revised manuscript, we have emphasized the novelty of our findings by highlighting as follows on page 24:

Importantly, our findings reveal a previously unrecognized role of TRPC6 as a direct regulator of mitochondrial homeostasis in the failing heart. By linking TRPC6 activity to both cytosolic signaling and mitochondrial function, this study identifies TRPC6 as a dual-compartment modulator of cardiac pathophysiology. This dual role not only expands the current understanding of ion channel biology in HFpEF but also highlights TRPC6 as a uniquely attractive therapeutic target.

Comment 5: Compensatory mechanisms resulting from the global KO, particularly from other TRPC channels such as TRPC3, may exist. The authors are encouraged to provide additional data, even in supplementary material, regarding the upregulation of other TRPC channels, particularly in the heart or mitochondria. This should be noted as a limitation or an avenue for future research.

Response: Our data showed that TRPC3 was not upregulated in TRPC6 KO mice at baseline and after HFD+L-NAME treatment in either cytosol or mitochondria (Figures 7A and C).  

Comment 6: Throughout the manuscript, ultrasound-based echocardiography was employed to illustrate the HEpEF; however, the authors did not incorporate specific cine loops. It is advised that the authors include cine loops for each result.

Response: In the revised manuscript, we included representative cine loops (echocardiographic videos) as supplementary material to illustrate the key echocardiographic findings supporting the HFpEF phenotypes. Specifically, we now provide cine loops comparing WT HFD+L-NAME and TRPC6 KO HFD+L-NAME mice after dobutamine treatment. These have been added as Supplementary videos 1 and 2 and are referenced accordingly in the Results section on page 14.

Comment 7: The authors are encouraged to include data that represent hemodynamic changes, specifically systolic and diastolic blood pressure.

Response: We included the blood pressure data measured by telemetry in our revision, including mean arterial BP and heart rate in Table 1.

Reviewer 4 Report

Comments and Suggestions for Authors

This study by Li, Fu, Dai et al. evaluates the role of TRPC6 in the development of heart failure with preserved ejection fraction (HFpEF), focusing on mitochondrial function, diastolic dysfunction, and cardiac reserve using a high-fat diet (HFD) and L-NAME-induced HFpEF mouse model. Authors proposed that TRPC6 contributes to mitochondrial dysfunction and cardiac impairment in HFpEF. TRPC6 knockout (KO) mice are protected from diastolic dysfunction, exhibit improved mitochondrial respiration, and reduced reactive oxygen species (ROS) generation, suggesting TRPC6 inhibition as a therapeutic strategy. The results are novel in terms of the role of TRPC6 in mitochondrial dysfunction, however several key points are missing to address:

Main comments:

  1. One of the main roles of TRPC6 in pathophysiology is to promote fibrosis. Here authors claim a trend to increase fibrosis but not reaching significance. This may reflect underpowered analysis (n=6).
  2. Another main points of the paper is the role of TRPC6 in mitochondria. First, the western results showed in supplementary figure one shows a vague band in the heart mitochondria, much lower than in the brain. Since this is the main point of the study it should be further confirmed using other methodologies like immunofluorescence or others.
  3. Where is TRPC6 expressed? In which type of cells? Literature suggests its role in fibrosis- then fibroblasts. What about cardiomyocytes? This is related to the model of TRPC6 KO. Authors state it as a general model, which raises the concern where does TRPC6 was knocked out. Since this channel is highly expressed in the brain and other tissues, do the mice have other side effects except the heart?
  4. Authors proposed that TRPC6 channel activity affected the Ca2+ in mitochondria. There is no direct evidence of TRPC6 channel activity on mitochondrial Ca²⁺ influx or ROS production.
  5. Additionally, HFpEF is more prevalent in women, however, here authors only used male mice. Please clarify and provide some limitations to the study.
  6. The timeline of the model is complex and the description is not clear. L-NAME administered during "the final 8 weeks" of a 12-week HFD, but methods lack clarity on whether HFD and L-NAME were concurrent or sequential. Please clarify or provide a schematic of how this model was produced.  Authors compared the severity of their model with Schiattarella et al., where a similar treatment produced a more severe phenotype. Please discuss in details why does your model is having a very moderate effect on cardiac physiology as compared to similar models in other laboratories.
Comments on the Quality of English Language

Overall the English quality is fine, however there are several comments

  1. During manuscript authors are using inconsistent and redundant phrasing. This needs to be rephrased to clarify the findings.
  2. There are several abbreviations during the manuscript that were not first introduced in the text. Please check that all of them are described in the main text.

Author Response

Reviewer #4:

Comment 1 One of the main roles of TRPC6 in pathophysiology is to promote fibrosis. Here authors claim a trend to increase fibrosis but not reaching significance. This may reflect underpowered analysis (n=6).

Response: In our study, fibrosis was assessed by Masson’s trichrome staining, and no significant differences were observed between groups. This likely reflects the early stage of HFpEF in our model, where diastolic dysfunction, hypertrophy, mitochondrial and metabolic changes, and endothelial dysfunction are evident, but overt fibrosis—typically a later event—has not yet developed. We clarified in the revised Discussion on page 22 that fibrosis may become more pronounced at later stages and that future studies with larger cohorts and longer follow-up will be important to define the role of TRPC6 in myocardial fibrosis.

In our model, fibrosis was not significantly detected by Masson’s trichrome staining. This likely reflects the relatively early stage of HFpEF being studied, where diastolic dysfunction, cardiomyocyte hypertrophy, mitochondrial and metabolic alterations are already evident, whereas interstitial and perivascular fibrosis typically emerge at later disease stages. It is also possible that our sample size limited the detection of more subtle fibrotic changes. Future studies with larger cohorts and extended time points will be important to determine whether TRPC6 contributes to the development of myocardial fibrosis as HFpEF progresses.”

Comment 2: Another main points of the paper is the role of TRPC6 in mitochondria. First, the western results showed in supplementary figure one shows a vague band in the heart mitochondria, much lower than in the brain. Since this is the main point of the study it should be further confirmed using other methodologies like immunofluorescence or others.

Response: We acknowledge that the western blot of TRPC6 in isolated cardiac mitochondria (Supplementary Figure 1) shows a relatively faint band compared with brain mitochondria. This difference reflects the lower abundance of TRPC6 in cardiac mitochondrial fractions relative to brain tissue. Importantly, the specificity of the band was confirmed by molecular weight consistency and the use of knockout tissue and multiple TRPC6 antibodies in our previous studies.

We agree that complementary approaches such as immunofluorescence or immunogold electron microscopy would provide stronger spatial confirmation of TRPC6 localization within mitochondria. While these experiments are beyond the current scope of this study, we added the paragraph below as a limitation and an important future direction in the revised Discussion on page 23.

Although our western blotting supports the presence of TRPC6 in cardiac mitochondrial fractions, the relatively lower abundance of TRPC6 in cardiac mitochondria compared to brain mitochondria and the technical limitations of fractionation warrant further validation using complementary methods such as immunofluorescence or high-resolution electron microscopy imaging. Future studies will be directed toward applying these approaches to more definitively establish mitochondrial localization of TRPC6 in the heart.

Comment 3: Where is TRPC6 expressed? In which type of cells? Literature suggests its role in fibrosis- then fibroblasts. What about cardiomyocytes? This is related to the model of TRPC6 KO. Authors state it as a general model, which raises the concern where does TRPC6 was knocked out. Since this channel is highly expressed in the brain and other tissues, do the mice have other side effects except the heart?

Response: TRPC6 is expressed in multiple tissues and cell types, including podocytes, neurons, endothelial cells, fibroblasts, and cardiomyocytes. In the context of cardiac remodeling, prior studies have emphasized its role in fibroblasts and fibrosis, but several reports, including our own, have also demonstrated TRPC6 expression in cardiomyocytes and its contribution to cardiomyocyte signaling pathways.

In the present study, we used a global TRPC6 knockout (KO) model and have acknowledged this limitation with the following discussion on page 23.   

“One limitation of the present work is the use of global TRPC6 knockout mice. TRPC6 is expressed in multiple tissues, including neurons, podocytes, endothelial cells, fibroblasts, and cardiomyocytes. We previously reported that global TRPC6 deletion alters systemic metabolism, leading to increased body weight and reduced leptin sensitivity (Am J Physiol Regul Integr Comp Physiol. 2022 Jul 1;323(1):R81-R97). Therefore, we cannot completely exclude systemic influences outside the heart. However, our current findings, together with ongoing studies in cardiac-specific TRPC6 knockout mice, support a direct role for TRPC6 in cardiomyocyte and cardiac remodeling processes in HFpEF.

Comment 4: Authors proposed that TRPC6 channel activity affected the Ca2+ in mitochondria. There is no direct evidence of TRPC6 channel activity on mitochondrial Ca²⁺ influx or ROS production.

Response: We agree that there is currently no direct evidence linking TRPC6 channel activity to mitochondrial Ca²⁺ influx or ROS production. Our data show that TRPC6 expression is increased in cardiac mitochondrial fractions in HFpEF, and we speculate that this may alter mitochondrial Ca²⁺ signaling. However, the relationship between TRPC6 and the mitochondrial calcium uniporter (MCU) remains unclear, and further studies will be required to determine whether TRPC6 directly contributes to mitochondrial Ca²⁺ homeostasis.

In our current study, we observed that mitochondria-generated ROS were elevated in isolated cardiac fibers from HFpEF mice. Whether this increase results from TRPC6 upregulation or secondary changes in mitochondrial Ca²⁺ handling is not yet known. We added these points to the revised Discussion section on page 23 and highlighted them as a key future direction for our research.

Another important future direction is to determine whether TRPC6 deficiency influences mitochondrial calcium handling through compensatory changes in MCU activity. While the precise interaction between TRPC6 and MCU remains unclear, prior studies have shown that genetic deletion of MCU alone does not confer protection against heart failure [48]. This highlights that mitochondrial calcium regulation in heart failure is multifaceted and likely involves additional pathways beyond MCU, warranting further exploration into the broader network of calcium-handling proteins.

Comment 5: Additionally, HFpEF is more prevalent in women, however, here authors only used male mice. Please clarify and provide some limitations to the study.

Response: We agree that the use of only male mice is a limitation of the present study. Several studies have reported that in the C57BL/6 HFpEF model induced by HFD+L-NAME, female mice are relatively protected compared to males. This protection has been attributed, at least in part, to the cardioprotective effects of estrogen at younger ages.

In our ongoing work, we have begun to investigate sex differences in aged mice. Interestingly, in older male and female mice, we observed the development of HFpEF in both sexes, suggesting that age-related loss of estrogen’s protective effects may contribute to sex convergence in disease susceptibility.

We added these points to the revised Discussion section on page 23, and we acknowledge that incorporating sex-based comparisons is an important future direction for elucidating the role of TRPC6 signaling in HFpEF.

“An important limitation of the present study is that it uses only male mice. Prior studies have shown that in the C57BL/6 HFpEF model induced by HFD+L-NAME, female mice are relatively protected compared to males, likely due to the cardioprotective effects of estrogen at younger ages. In our ongoing work, we have begun to address this by examining aged cohorts, where we observed HFpEF development in both sexes, suggesting that age-related loss of estrogen’s protective effects may contribute to sex convergence in disease susceptibility. Future studies incorporating sex-based comparisons will be essential to fully define the role of TRPC6 signaling in HFpEF pathogenesis.

Comment 6: The timeline of the model is complex and the description is not clear. L-NAME administered during "the final 8 weeks" of a 12-week HFD, but methods lack clarity on whether HFD and L-NAME were concurrent or sequential. Please clarify or provide a schematic of how this model was produced.  Authors compared the severity of their model with Schiattarella et al., where a similar treatment produced a more severe phenotype. Please discuss in details why does your model is having a very moderate effect on cardiac physiology as compared to similar models in other laboratories.

In our study, mice were fed a 45% HFD for 4 weeks, followed by concurrent HFD and L-NAME treatment for an additional 8 weeks. We have added a schematic Figure 1 to clearly illustrate this timeline in the revised Methods.

When comparing our model to that of Schiattarella et al., there are two key differences that may explain the more moderate phenotype in our study:

  1. Diet composition: Schiattarella et al. employed a 60% HFD combined with L-NAME for 5 or 15 weeks. By contrast, we used a 45% HFD for 12 weeks, which more closely approximates a Western-style human diet. The mice were treated with LNAME for 8 weeks. This may explain the milder phenotype in our mice and enhances the clinical relevance of our model for studying earlier HFpEF stages.
  2. Mouse strain background: Schiattarella et al. used C57BL/6N mice, whereas we employed a B6/129s mixed background. It is well known that genetic background significantly influences cardiovascular remodeling and metabolic responses, which may also contribute to differences in phenotype severity between the models.

We included these points in the revised Discussion on page 21 to provide potential reasons why our HFpEF phenotype appears more moderate compared with the study of Schiattarella et al.

 “Our results differ somewhat from Schiattarella et al.'s study [20], which used a 60% HFD combined with L-NAME for 15 weeks in C57BL/6M mice. Although similar pathological phenotypes of HFpEF, including cardiac hypertrophy, were observed in both studies, we did not observe significant cardiac fibrosis after HFD+L-NAME in WT or TRPC6 KO mice. This difference may be due to differences in the length of time for L-NAME treatment, the fat content in the diets, and different strains of mice. However, the primary phenotypes of HFpEF, including cardiac hypertrophy, preserved EF, impaired diastolic function, and reduced exercise tolerance, were similar in both studies.

Comment 7: During manuscript authors are using inconsistent and redundant phrasing. This needs to be rephrased to clarify the findings.

Response: In the revised version, we carefully rephrased and streamlined the text throughout the Abstract, Results, and Discussion to ensure consistent terminology and to present the findings more clearly.

Comment 8: There are several abbreviations during the manuscript that were not first introduced in the text. Please check that all of them are described in the main text.

Response: We carefully reviewed the manuscript and defined abbreviations upon first use. 

Reviewer 5 Report

Comments and Suggestions for Authors

The article refers to the attenuation of cardiac and mitochondrial dysfunction in heart failure with preserved ejection fraction induced by a high-fat diet plus L-NAME.

The summary is consistent with the article, although it is necessary to evaluate whether the correct word is protect or prevent, since the TRPC6 deficiency is due to KO, so it may not be feasible to use the word protect; in that case, partially prevents would be more appropriate.

The results are clear, although it is necessary to define the term moderate in some results.

In the discussion, it is necessary to expand on the explanation, because only mentioning time as a reason for the change makes it unclear whether it is due to accumulation or the effect of L-NAME at the central level.

It is necessary to give an explanation of the mechanisms involved in the changes in TRPC6 KO expression.

The conclusion is very advanced in relation to the results obtained, because mentioning that it is a strategy to improve mitochondrial function when TRPC6 KO is used does not make it clear what such a strategy would be.

Author Response

Reviewer #5:

Comment 1: The summary is consistent with the article, although it is necessary to evaluate whether the correct word is protect or prevent, since the TRPC6 deficiency is due to KO, so it may not be feasible to use the word protect; in that case, partially prevents would be more appropriate.

Response: We have changed the “protect” to “partially prevents” per your suggestion.

Comment 2: The results are clear, although it is necessary to define the term moderate in some results.

Response: In the revised manuscript, we clarified the term moderate when describing certain results. Specifically, we now provide more precise descriptions as a percent difference to control to ensure that the reader can clearly understand what " moderate " means in the context of our findings.

Comment 3: In the discussion, it is necessary to expand on the explanation, because only mentioning time as a reason for the change makes it unclear whether it is due to accumulation or the effect of L-NAME at the central level.

Response: In the revised Discussion, we expanded our explanation beyond “time” as the only factor. Specifically, we now clarify that the observed changes may reflect both cumulative effects of prolonged metabolic and hemodynamic stress and the direct impact of chronic L-NAME administration, which can influence vascular tone and nitric oxide signaling at both the peripheral and central levels. By distinguishing between these possibilities, we provide a more nuanced interpretation of the mechanisms that may underlie the progression of HFpEF in our model, as shown below on page 20.

The progression of HFpEF in our model may not be explained solely by the duration of exposure but also by the accumulated metabolic stress from the high-fat diet in combination with the sustained inhibition of nitric oxide synthase by L-NAME. While prolonged exposure likely contributes to the gradual worsening of cardiac function, chronic L-NAME treatment may exert additional effects on vascular regulation, endothelial dysfunction, and possibly central mechanisms of blood pressure control. Future studies will be needed to dissect the relative contributions of these cumulative effects.

Comment 4: It is necessary to give an explanation of the mechanisms involved in the changes in TRPC6 KO expression.

Response: In the revised Discussion shown below on page 23, we incorporated mechanistic considerations and identified them as important future directions for more targeted studies using cell-specific TRPC6 knockout models.

“The changes observed in TRPC6 KO mice likely arise from altered Ca²-dependent signaling pathways. TRPC6 activates calcineurin–NFAT signaling and contributes to cardiomyocyte hypertrophy and fibroblast-mediated fibrosis. Reductions of TRPC6 activity may therefore blunt these maladaptive responses. In addition, TRPC6 may influence mitochondrial Ca² homeostasis and ROS generation, linking channel activity to mitochondrial dysfunction in HFpEF. Because TRPC6 is expressed in cardiomyocytes, fibroblasts, and endothelial cells, systemic deletion may affect intercellular communication within the myocardium. These mechanisms together provide a rationale for the phenotypic changes observed in TRPC6 KO mice, though further cell-specific studies will be required to fully define these pathways.

Comment 5: The conclusion is very advanced in relation to the results obtained, because mentioning that it is a strategy to improve mitochondrial function when TRPC6 KO is used does not make it clear what such a strategy would be.

Response: We agree that the TRPC6 knockout should not be represented as a direct therapeutic strategy. Our results present a mechanistic link rather than a defined therapeutic strategy. We have clarified this distinction in the revised Conclusion and Discussion as follows on page 24.

While our results do not establish TRPC6 deficiency as a therapeutic strategy, they provide new mechanistic insight and suggest that modulating TRPC6 activity could represent a potential avenue for therapeutic intervention in HFpEF.

Round 2

Reviewer 1 Report

Comments and Suggestions for Authors

Authors have satisfactorily addressed comments.

Author Response

Authors have satisfactorily addressed comments.

Reviewer 2 Report

Comments and Suggestions for Authors

The authors have responded appropriately to the comments. I have nothing else to add.

Author Response

The authors have responded appropriately to the comments. I have nothing else to add.

Reviewer 4 Report

Comments and Suggestions for Authors The revised manuscript by Li et al. shows significant improvement and addresses most of the concerns raised in the previous review. The authors have successfully enhanced the clarity of the results and discussion sections, and their rationale for the observed phenotype is now more compelling. Specifically, the explanations for the lack of fibrosis and the use of a global knockout model are well-reasoned and scientifically sound. Additionally, the conclusions have been strengthened, providing a clear and concise summary of the key findings regarding TRPC6's role in HFpEF. However, there are still a few remaining issues that need to be addressed. While the authors have added discussion regarding the faint western blot band for mitochondrial TRPC6, this point remains a minor weakness. A more detailed acknowledgment of the semi-quantitative nature of this finding in the main text would enhance the manuscript. This aspect is important for contextualizing the claims about TRPC6's mitochondrial localization. Overall, the manuscript is much improved and is approaching publishable quality. With some additional detail on the remaining points, particularly concerning the interpretation of the western blot data, I believe the paper will be ready for publication. I suggest a minor revision to address these final issues.

Author Response

Comment 1: However, there are still a few remaining issues that need to be addressed. While the authors have added discussion regarding the faint western blot band for mitochondrial TRPC6, this point remains a minor weakness. A more detailed acknowledgment of the semi-quantitative nature of this finding in the main text would enhance the manuscript. This aspect is important for contextualizing the claims about TRPC6's mitochondrial localization.

Response 1: We appreciate the reviewer’s thoughtful point. We agree that the mitochondrial TRPC6 western blot signal is faint and that subcellular fractionation–based immunoblotting is inherently semi-quantitative. To address this, we added the following statement in the main text on line 234: “Because the TRPC6 signal detected in mitochondrial fractions is low in intensity and subcellular fractionation is inherently semi-quantitative, we interpret these immunoblots as evidence of increased enrichment of TRPC6 immunoreactivity in the mitochondrial fraction after HFD+L-NAME rather than definitive proof of intramitochondrial localization. Densitometry is reported as TRPC6 normalized to VDAC and expressed relative to WT-RD.”,

We also have a statement previously added in the limitation of discussion on line 420: “Second, although our western blotting supports the presence of TRPC6 in cardiac mitochondrial fractions, the relatively lower abundance of TRPC6 in cardiac mitochondria compared to brain mitochondria and the technical limitations of fractionation warrant further validation using complementary methods such as immunofluorescence or high-resolution electron microscopy imaging. Future studies will be directed toward applying these approaches to more definitively establish mitochondrial localization of TRPC6 in the heart.”